# Nutritional Counseling and Mediterranean Diet in Adrenoleukodystrophy: A Real-Life Experience

**DOI:** 10.3390/nu16193341

**Published:** 2024-10-01

**Authors:** Maria Rita Spreghini, Nicoletta Gianni, Tommaso Todisco, Cristiano Rizzo, Marco Cappa, Melania Manco

**Affiliations:** 1UOC of Endocrinology and Diabetology, Bambino Gesù Children’s Hospital, Istituti di Ricovero e Cura a Carattere Scientifico, IRCCS, 00165 Rome, Italy; mariarita.spreghini@opbg.net (M.R.S.); tommaso.todisco@chuv.ch (T.T.); 2Research Unit for Predictive and Preventive Medicine, Bambino Gesù Children’s Hospital, Istituti di Ricovero e Cura a Carattere Scientifico, IRCCS, 00165 Rome, Italy; nicoletta.gianni@opbg.net; 3Service of Endocrinology, Diabetology and Metabolism, Lausanne University Hospital, 1011 Lausanne, Switzerland; 4UOC of Metabolic Diseases, Bambino Gesù Children’s Hospital, Istituti di Ricovero e Cura a Carattere Scientifico, IRCCS, 00165 Rome, Italy; cristiano.rizzo@opbg.net; 5Research Unit for Innovative Therapies for Endocrinopathies, Bambino Gesù Children’s Hospital, Istituti di Ricovero e Cura a Carattere Scientifico, IRCCS, 00165 Rome, Italy; marco.cappa@opbg.net

**Keywords:** adrenoleukodystrophy, X-ALD, hexacosanoic acid, C26:0, Mediterranean diet, nutritional therapy, VLCFA-restricted diet, very-long-chain fatty acids, VLCFAs

## Abstract

**Background/Objectives**: Adrenoleukodystrophy (X-ALD) is a metabolic disorder caused by dysfunctional peroxisomal beta-oxidation of very-long-chain fatty acids (VLCFAs). A VLCFA-restricted Mediterranean diet has been proposed for patients and carriers to reduce daily VLCFA intake. **Methods**: We retrospectively evaluated plasma VLCFAs in a cohort of 36 patients and 20 carriers at baseline and after 1 year of restricted diet. **Results**: At T1, compliant adult patients had significantly lower C26:0 levels [1.7 (1.2) vs. 2.5 µmol/L (1.7), *p* < 0.05], C26:0/C22:0 ratio [0.04 (0.02) vs. 0.06 (0.03), *p* < 0.05], and triglycerides [93 (56.5) vs. 128 mg/dL (109.5), *p* < 0.05] than non-compliant ones. C26:0 [2.4 (1.7) vs. 1.7 (1.2) µmol/L, *p* < 0.05], the C26:0/C22:0 ratio [0.06 (0.04) vs. 0.04 (0.02), *p* < 0.05], and cholesterol [173.5 (68.3) mg/dL vs. 157 (54) mg/dL, *p* < 0.05] were significantly reduced in compliant adult patients at T1 vs. baseline. As for carriers, the C26:0/C22:0 ratio was lower [0.02 (0.01) vs. 0.04 (0.009), *p* < 0.05] at T1 in compliant carriers, as compared to non-compliant ones. The C26:0/C22:0 [0.03 (0.02) vs. 0.02 (0.01) *p* < 0.05] and C24:0/C22:0 [1.0 (0.2) vs. 0.9 (0.3), *p* < 0.05] ratios were significantly decreased at T1 vs. T0. **Conclusions:** A VLCFA-restricted diet is effective in reducing plasma VLCFA levels and their ratios and must be strongly encouraged as support to therapy.

## 1. Introduction

Adrenoleukodystrophy (X-ALD, 300100) [1], a rare metabolic disorder with estimated incidence at 1/20,000 [2], causes dysfunction in peroxisomal beta-oxidation and, consequently, increased plasma levels of very-long-chain fatty acids (VLCFAs, longer than C22). X-ALD is determined by a mutation in the ABCD1 gene, on chromosome X (Xq28), encoding Adrenoleukodystrophy Protein (ALDP), a member of the ATP-binding cassette transport family, carrying VLCFAs into peroxisomes. More than 600 mutations have been identified to date, thus partially explaining variability in the clinical manifestation of the disease. VLCFAs accumulate in the central nervous system, adrenal cortex, and gonads, leading to adrenal dysfunction and central and peripheral demyelination [3,4].

The clinical features of X-ALD phenotypes are described in Appendix A; the early onset of pediatric X-linked Cerebral ALD (CerALD) and Adrenomyeloneuropathy (AMN) are the prevalent phenotypes, accounting for 48% and 25% of patients, respectively. Male patients carrying a gene defect without clinical signs are referred to as presymptomatic individuals, being at increased risk of developing the overt disease [3,5,6,7,8,9]. Women, who are gene carriers, are symptomatic after the fourth decade with a mild form of the disease, presenting ataxia, leg pain, incontinence, mild myelopathy, or, rarely, severe myelopathy [3,4,9,10].

The loss of function of ALDP correlates with high plasma levels of VLCFAs in the brain and adrenal glands, as revealed by postmortem analysis [3,5,6]. VLCFAs are esterified to cholesterol, glycerophospholipids, phospholipids, and sphingolipids, likely triggering an impairment in cerebral white matter and changes in the integrity of myelin, with demyelination and a reduction in cerebrosides, and moreover determining an increased inflammation and oxidative stress, owing to increased levels of reactive oxygen species. Dyslipidemia with high levels of total cholesterol is due to the dysfunctional transport and metabolism of cholesterol esterified to VLCFAs and to the reduced availability of cholesterol for the synthesis of steroid hormones. In adrenal glands, the excess of VLCFAs determines an alteration of membrane microviscosity, leading to an attenuated response to the adrenocorticotropic hormone and hypoadrenalism [4,6,7,11].

Only a small fraction of VLCFAs derive from the diet, while the majority are synthetized endogenously by the elongase enzyme ELOVL1 (ELOngation of Very-Long-chain fatty acids-1) [6]. Both C26:0 (hexacosanoic acid) and C24:0 (tetracosanoic acid), and their ratios to C22:0 (docosanoic acid), are biomarkers for suspected disease. Their measurements are recommended for the following groups: men with non-autoimmune hypoadrenalism, with or without gait and/or behavioral disturbances; men with behavioral disturbances and areas of demyelination identified by magnetic resonance imaging; men with progressive paraparesis; and women with idiopathic progressive paraparesis and/or positive family history. The suspected disorder should be confirmed by a genetic test that identifies the mutated gene [3,7].

Neonatal screening allows for a more accurate estimate of the prevalence and early identification [3,12], this being pivotal for therapeutic intervention, as well as a VLCFA-restricted diet to reduce deleterious effects and alleviate symptoms, delay the onset of symptoms, and improve the prognosis in pre-symptomatic patients and carriers.

In this retrospective study, we aimed at investigating the impact of a real-life nutritional scenario with the effect of reducing plasma VLCFA levels with a restricted Mediterranean diet (the “Bambino” diet), in patients with X-ALD and female carriers.

## 2. Materials and Methods

To evaluate the efficacy of the VLCFA-restricted nutritional treatment, we retrospectively reviewed clinical and biochemical data from Electronic Health Records (EHRs) of all the patients affected by either Cer-ALD or AMN, as well as Addison-only patients and female carriers, referred to the Unit of Endocrinology and Diabetology of the Bambino Gesù Children’s Hospital (OPBG, Ospedale Pediatrico Bambino Gesù), from January 2019 to July 2023. We included 56 subjects in total, distributed as follows: 26 male adults, 10 male children, and 20 female adults.

Patients and carriers underwent a nutritional evaluation and were prescribed with the VLCFA-restricted Mediterranean diet (the “Bambino” VLCFA-restricted Mediterranean diet). Oil and butter could not be included in the restricted diet due to their high content of C26:0 and fats, and were replaced with Aldixyl OiLife, a special oil composed of pure oleic acid (Aldixyl OiLife; Pharmaelle, Bologna, Italy) and administered as needed, with a suggested dose of 1 spoon/day, as part of the nutritional treatment.

All the patients were prescribed with a food for special medical purposes, consisting of a mixture of trioleic and trierucic acids in a ratio of 4:1, conjugated linoleic acids, and antioxidants (Aldixyl; Pharmaelle, Bologna, Italy) [13,14]. The prescribed dose was 0.25 to 0.75 mL/kg/day (a lower dose, 15 to 20 mL/day, was usually administered in pediatric patients).

The study baseline (T0) was the date of the restricted diet prescription, and T1 the follow-up visit after 1 year. Plasma VLCFA levels (C26:0, C26:0/C22:0 ratio, and C24:0/C22:0 ratio), total cholesterol, and triglycerides were measured at T0 and T1. Compliance to nutritional treatment was evaluated with a 24 h Recall, compiled by dietitians (MRS and NG) during the in-person follow-up visit and with a 7-day Food Diary, compiled by the patient.

### 2.1. Anthropometrics and Biochemical Evaluation

Height was measured using a wall Holtain-stadiometer, and weight was measured with scales certified for medical use (90/384/EEC, SECA, Hamburg, Germany). Patients wore minimal clothing and no shoes during these measurements. The average of two measurements was used to calculate the Body Mass Index (BMI). All measures were taken to ensure the confidentiality of participants whose data were used. 

Blood samples were collected in EDTA tubes, and plasma was stored at −20 °C for up to six months. Plasma VLCFAs (from C22:0 to C26:0) were identified and quantified by gas chromatography/mass spectrometry, after extraction and derivatization to methyl esters, using a control plasma, certified ERNDIM (European Research Network for evaluation and improvement of screening, Diagnosis and treatment of Inherited disorders of Metabolism). Plasma C26:0 levels are expressed in µmol/L. C26:0/C22:0 ratio (hexacosanoic to docosanoic acid) and C24:0/C22:0 ratio (tetracosanoic to docosanoic acid) were then calculated, being crucial for therapeutic monitoring and diagnosis of X-ALD in carriers. The normal ranges are the following: hexacosanoic acid 0.010–0.900 µmol/L, C26:0/C22:0 ratio 0.006–0.020, and C24:0/C22:0 ratio 0.470–1.270 [15]. 

Total cholesterol and triglyceride levels were assessed using colorimetric kits (modular systems P/S Can 433; Roche Diagnostic/Hitachi, Monza, Italy).

### 2.2. The “Bambino” VLCFA-Restricted Mediterranean Diet

We specifically designed the “Bambino” diet to limit VLCFA intake to less than 3 mg/day (the lowest threshold identified in Van Duyn et al.’s study [16]), while minimizing total fatty acid consumption. Due to the lack of national or international reference data for C26:0 content in foods, we relied on C26:0 content values reported in the studies by Van Duyn et al. (1984) [16] and Kawahara et al. (1988) [17]. We set the C26:0 cut-off at 0.150 mg/100 g serving, excluding any foods with C26:0 content equal to or higher than this threshold [16,17]. Both studies referred to gross weight or baked weight, depending on the food. 

We also set a maximum fat content limit of 2 g/100 g of the edible portion. We excluded foods with fat content equal to or higher than the cut-off, as reported in Italian or US databases [18,19,20]. For foods not listed in the CREA (Consiglio per la Ricerca in Agricoltura e l’Analisi dell’Economia Agraria, 2019) database, we referred to either the BDA (Banca Dati di Composizione degli Alimenti per Studi Epidemiologici in Italia, 2022) or the USDA (U.S. Department of Agriculture, 2019) databases, particularly for items like spices [18,19,20]. 

The food categories we analyzed included cereals, meat and meat-derived products, fish, legumes, eggs, milk and yogurt, dairy products, fruits, vegetables, spices, nuts and seeds, fried foods, pre-packaged and precooked foods, sweets, beverages, and seasonings. When discrepancies among the databases were observed, we considered the highest value and excluded the product if it met or exceeded the set limits. 

In our food analysis, we adopted a non-brand-specific scenario. Many of the brand names referenced in the US and Japanese studies by Van Duyn et al. and Kawahara et al. [16,17] are not available in Italy. As a result, we considered only the available nutritional value, without considering or comparing the brand. Nutritional variations between Italian brands were not considered; instead, we used the average nutritional values from non-branded items as reported in food composition databases. Neither were differences in nutrient content due to agriculture, cultivation, and environmental factors considered. 

After a thorough analysis, we selected foods based on the Mediterranean dietary pattern, which aligns with the eating habits of most of our patients and their families, as reported in their Food Diaries. We categorized these foods into three groups: non-allowed foods, foods with hidden fats, and allowed foods. We listed the foods in Appendix A. Among “non-allowed foods” group, we included both category products with elevated levels of C26:0 (≥0.150 mg/100 g) and the ones with high content of fats (≥ 2 g/100 g) (Appendix A). The “foods with hidden fats” category contains items that appear not to have high fat or VLCFA content, but are made from forbidden ingredients such as eggs, cheese, olive oil, milk, and some sausages (Appendix A). The “allowed foods” group consists of items all featuring low levels of fatty acids and/or low amount of VLCFAs, including non-whole-grain cereals, lean cuts of meat, certain types of fish, and legumes (peeled when possible). Foods made with skim milk (0% fat) are also included. Fruits, vegetables, spices, and beverages not mentioned in the other categories are permitted, with regard to the peel and seeds of plant foods, which must be removed before eating, as they can be sources of C26:0 [16,21] (Appendix A). Although white bread, rice, and other white flour products are relatively high in C26:0, we categorized them as “allowed foods” because they play an important role in the Mediterranean diet. By allowing their consumption, we ensure an adequate variety of dietary pattern and a balanced intake of all the nutrients, making the restricted diet more feasible for patients and carriers. Replacing these foods with alternative cereals or pseudocereals was not reasonable, as these alternatives contain higher levels of fatty acids.

### 2.3. Evaluation of Adherence to the Diet

Patients were provided with lists of allowed and non-allowed foods, along with general advice on consuming certain items, to ensure the best compliance to the diet. Rather than prescribing specific portions, patients were educated to make appropriate food choices based on their physiological needs and their hunger and satiety cues, conforming to the nutritional education therapy.

At the follow-ups, all patients received nutritional counseling. We assessed their adherence to the dietary protocol using two methods: the 24-h Recall and the Food Diary. The 24-h Recall is an assessment tool based on a short interview performed by the operator, in which the participants recall foods and drinks they consumed in the previous 24 h. The Food Diary is a 3- or 7-day assessment tool, self-administered by subjects, and records all meals eaten during that period. It is potentially affected by a minor risk of bias, such as inaccuracies due to false memories. Both tools aim at evaluating patients’ eating habits and their compliance with the prescribed nutritional therapy. 

### 2.4. Statistical Analysis

Data were reported as median and Interquartile Range (IQR). We used a non-parametric based-ranks test to assess the significant differences between groups. Comparisons were carried out using the Wilcoxon test. A *p*-value at 5% was considered statistically significant. Data analysis was performed using the R for Windows statistical software, version 3.0.3.

## 3. Results

A total of 78 EHRs were re-evaluated, with data from 56 subjects—36 affected males and 20 female carriers—ultimately included in the analysis. We excluded patients with severe dysphagia requiring enteral nutrition (*n* = 8) and those who did not complete the follow-up at T1 (*n* = 14). The studied cohort was aged from 4.8 to 72 years [men: 39.1 (22.3); children: 8.9 (8.1); carriers: 51.7 (14.3)]. At the follow-up (T1), participants were re-evaluated after a median period of 13 months (range: 8 to 33 months). Table 1 presents the anthropometric and biochemical measurements at the baseline (T0) and T1 for the three groups: adults, children, and female carriers.

Fifty percent of the adult and child patients adhered the nutritional advice (fourteen out of twenty-six adults and five out of ten children), while in carriers, we observed a higher percentage of adherence (*n* = 16; 80%).

Total C26:0 [1.7 (1.2) vs. 2.5 µmol/L (1.7), *p* < 0.05], C26:0/C22:0 ratio [0.04 (0.02) vs. 0.06 (0.03), *p* < 0.05] and triglycerides [93 (56.5) vs. 128 mg/dL (109.5) mg/dL, *p* < 0.05] at T1 were significantly lower in compliant adult patients as compared to non-compliant ones (Figure 1A–C). Total C26:0 [2.4 (1.7) vs. 1.7 (1.2) *p* < 0.05], C26:0/C22:0 ratio [0.06 (0.04) vs. 0.04 (0.02) *p* < 0.05] and total cholesterol [173.5 mg/dL (68.3) vs. 157 mg/dL (54), *p* < 0.05] were significantly decreased at T1 as compared to baseline in compliant individuals (Figure 2A–C).

In the group of child patients, we did not find a significant difference in C26:0 levels or in the ratio following the nutritional intervention, regardless of the treatment compliance.

In the 20 carriers, there was a significant reduction in C26:0/C22:0 ratio [0.02 (0.01) vs. 0.04 (0.009), *p* < 0.05] at T1 in those who adhered to the nutritional treatment as compared to those who did not (Figure 3), and in the C26:0/C22:0 ratio [0.03 (0.02) vs. 0.02 (0.01), *p* < 0.05] and C24:0/C22:0 ratio [1.0 (0.2) vs. 0.9 (0.3), *p* < 0.05], comparing longitudinal values at T0 versus those at T1 (Figure 4A,B).

Analysis of patients’ and carriers’ Food Diaries demonstrated a drastic average reduction in C26:0 intake from 12.1 mg/day to 2.5 mg/day, with the same caloric content of about 1400–1500 kcal/day distributed over five meals per day in those who adhered to the restricted diets as compared with those who did not adhere. Total fatty acid intake decreased from about 20% to about 6% with daily oil seasoning not included (Table 2).

## 4. Discussion

The “Bambino” diet prescribed in our study is designed on a Mediterranean pattern, with special attention to those foods high in C26:0 and fats. While some of the pillars of the Mediterranean diet are missing in the “Bambino” diet, it maintains a considerable variety. It still incorporates a wide range of fruits, vegetables, and legumes, with only a few exceptions, as well as cereals. Notably, bread, rice, and products made from white flour are included, although having high levels of VLCFAs, thus aligning with the Mediterranean diet’s principles and ensuring an adequate food variety and nutrient intake. Nevertheless, the consumption of extra virgin olive oil is excluded from the diet, being replaced by the X-ALD specifically formulated oil.

The “Bambino” diet restricted in VLCFAs leads to a significant improvement of biochemical markers of X-ALD disease, such as plasma C26:0 content and its ratios. A reduction in circulating levels of both biomarkers was evident in patients and carriers complying with the diet restrictions. 

We considered “non-compliant” patients the ones declaring not to follow general advice and/or eating non-allowed foods and/or using common seasoning oils. Analysis of the Food Diaries and 24-h Recalls indicated good adherence to the diet, with nearly 62.5% of participants being compliant. However, among both children and adults, the lowest compliance was registered, with a major consumption of non-allowed foods (industrial and processed items, such as sweets, meat-derived products, sweetened beverages), while joining parties and social events. Thus, recreational events were the major obstacle to adherence.

In adult patients who adhered to the diet, we found lower triglyceride levels compared to their non-compliant counterparts at T1. While, in the adult male group, total cholesterol was lower at T1, no differences were found in total cholesterol and triglycerides levels in carriers.

Given that a small portion of VLCFAs originate from exogenous sources [8,16,17,23], the “Bambino” VLCFA-restricted diet serves as an adjunctive treatment. It results in the significant restriction of C26:0 intake (up to 2.5 mg/day in the sample diet reported here). Importantly, we also observe a decrease in total fat intake (6%, with daily oil seasoning not included (see Table 2)). The adequate seasoning oil (Aldixyl OiLife), composed of pure oleic acid, is then part of the nutritional treatment, due to high levels of VLCFAs in common seasonings.

When comparing the food for a special medical purpose (Aldixyl; Pharmaelle, Bologna, Italy) to the first oil therapy, “Lorenzo’s Oil” [24,25], the mixture of triolein and trierucin in a 4:1 ratio, conjugated linoleic acids and antioxidants (alpha-lipoic acid, L-glutathione reduced, vitamin E), proposed by Cappa et al. [13,14], permits crossing of the blood–brain barrier. It has been demonstrated to inhibit the accumulation of VLCFAs, interfere with elongases to prevent synthesis of VLCFAs, and increase the activity of ALDP and consequently beta-oxidation in peroxisomes. Thanks to the addition of antioxidants, inflammation and oxidative stress are also reduced. In a two-month study involving five women with no or moderate symptoms, this newly formulated oil caused significant outcomes: a decrease in total cholesterol and LDL (Low-Density Lipoprotein) cholesterol, an increase in HDL (High-Density Lipoprotein) cholesterol, a decrease in plasma C26:0 levels and C26:0/C22:0 ratio, and an increase in the C22:6/C20:5 (docosahexaenoic acid/eicosapentaenoic acid) ratio, markers of peroxisomal beta-oxidation. Levels of interleukin-6 were significantly reduced in plasma and cerebrospinal fluid [13,14].

The average C26:0 intake in our study was comparable to levels reported in previous studies on restricted diets [16,17,23,26]. The diet described by Van Duyn et al. in 1984 [16] aimed to limit daily C26:0 intake to 3 mg, far below the usual intake of VLCFAs in the US diet [16]. A higher threshold of 10 mg/day was suggested by Kawahara et al. (1988) [17] and Rizzo et al. (1987) [26]. However, these studies were conducted with small sample sizes, with Van Duyn et al. including only seven patients [16]. Adherence to the diet and VLCFA intake restrictions were assessed using Food Diaries. This study did not find significant changes in plasma C26:0 levels, nor amelioration in the clinical prognosis of the disease [16]. Despite this, they claimed the potential benefit of dietary restriction of VLCFAs, combined with other therapies. Patients were followed for a time frame varying from 4 to 24 months and the possibility of poor adherence to the diet was mentioned by the authors as a plausible explanation for the lack of significant difference in C26:0 plasma level following the diet with respect to the baseline [16]. Another relevant study by Moser et al. [23] included 36 individuals and analyzed plasma VLCFA levels in patients following a diet restricted to 3 mg/day of C26:0, along with the administration of glyceryl trioleate oil, for a range period that varied from 60 days to 1.5 years. This cohort encompassed all the phenotypes of X-ALD and also included asymptomatic individuals and carriers. The authors revealed lower plasma VLCFA levels in 25 patients out of 36. They also observed improvement in peripheral nerve function in two individuals [23]. When considering the rarity of the disease, the “Bambino” study sample size appears to be larger, and our findings are in keeping with those of Moser et al.’s study (1987) [23].

Few studies addressing the putative role of VLCFA-restricted diets in the management of X-ALD demonstrate that these attempts are feasible for different dietary patterns, such as US and Japanese diets [16,17]. We worked on it, demonstrating that this attempt is also feasible using the Mediterranean diet, according to our lists of foods, based on C26:0 and total fat content. Given the lack of comprehensive data on food content of these nutrients in the major nutrient databases, we provided a ready-to-use list of allowed and non-allowed foods for patients, carriers, and caregivers, providing them with education on their consumption.

To build up these lists, we set the cut-off of C26:0 at 0.150 mg/100 g serving, following the guidelines by Van Duyn et al. (1984) [16] and Kawahara et al. (1988) [17], who are the only sources of information on VLCFA content in foods available in the scientific literature. They performed lipid analyses on food samples and documented C26:0 intake with the diet in the general populations of the US and Japan, reporting daily intakes of 15–40 mg and 12–36 mg, respectively [16,17]. Van Duyn et al. (1984) [16] found that in cereals, the higher the fiber content, the higher the C26:0 content was found to be; in particular, whole-grain products have the higher content, as compared to their non-whole-grain counterparts. They also identified elevated C26:0 levels in white bread, white flour, and homemade Italian bread [16]. Kawahara et al. (1988) [17] confirmed most of the findings from the earlier study. The two studies [16,17] differed slightly from each other, both confirming high levels in white bread and also detecting high levels in white rice. Among protein sources, fish, meat, legumes, and non-skimmed dairy products exhibited the highest VLCFA content. For fruits and vegetables, the C26:0 content per 100 g of edible portion was mostly low, except for the varieties listed in Appendix A. Notably, significant differences were observed between peeled and unpeeled fruits, with the outer coverings of plant foods containing the highest VLCFA levels. Therefore, the fruits and vegetables indicated in Appendix A should be avoided, even if unpeeled. Additionally, spices, lard, and oils are known to have extremely high levels of C26:0 [16,17,21]. Unfortunately, current freely available databases do not report C26:0 content, making it challenging to assess VLCFA levels in foods.

It is worth noting that Van Duyn et al. (1984) [16] also suggested limiting total fat intake per day. Indeed, when they compared an average US diet with a diet restricted to VLCFAs of the same caloric content, they observed that a large decrease in C26:0 intake was paralleled by a moderate decrease in total daily fat intake (from 31% to 12%). Thus, we excluded high-fat foods, defining “high-fat” as a fatty acid content greater than (or equal to) 2 g/100 g of edible portion, as resulted from the analysis of food composition databases of CREA, BDA, and USDA [18,19,20]. Looking at products not listed in the previous studies [16,17], we found that pseudocereals with higher fiber content, certain fatty meats, spices, nuts and seeds, fried foods, precooked industrial products, sweets, and various beverages had the highest fat content [18,19,20]. We found moderate discrepancies in fat content values between databases for some foods, possibly due to non-standard experimental procedures.

Due to the large exclusion of foods, as general nutritional advice, we strongly recommended to our patients to eat fresh foods and avoiding prolonged cooking, to ensure adequate intake of all micronutrients and limit the risk of vitamin and mineral deficits. 

The growing scientific interest in the nutritional management of X-ALD has been enhanced by the establishment of newly developed screening programs for X-ALD. Indeed, a nutritional intervention to reduce the VLCFA content in the diet may be beneficial to delay the onset of neurologic symptoms or improve them in patients and carriers. Neonatal screening for X-ALD should be an important tool to make the diagnosis earlier and consequently start nutritional intervention as soon as possible [27,28,29,30,31].

Our study is a retrospective but real-life investigation. Ethical concerns make it very complicated to carry out a randomized controlled trial in X-ALD, a rare condition [8], which also accounts for the small number of patients evaluated in the present study, despite our sample size being larger when compared to previous studies [16,23].

When studying nutritional treatment in X-ALD, a limit is caused by the lack of up-to-date national or international dietary tables that provide detailed information on the nutritional values of interest, first of all C26:0, in common Italian foods. Moreover, discrepancies in C26:0 content reported by previous studies [16,17] stem from the different techniques used for analysis. To support future research, it is crucial to standardize VLCFA measurement methods and analytical procedures to ensure consistent and comparable data. Despite this, the findings from the OPBG experience, along with the great compliance to the nutritional treatment shown by the enrolled individuals, has proven it possible to lower consumption by up to 2.5 mg/day, following our sample of a restricted diet, together with a total fat intake decrease too. The great amount of allowed foods according to the Mediterranean pattern makes it sustainable, as proven by data on adherence revealed by statistical analysis. However, more frequent in-person follow-up visits could have improved compliance. Age-specific behavioral strategies to comply with the diet must be elaborated in children. 

As a limit of our study, we did not analyze data on micronutrient intake, which could occur in patients following a restricted diet, due to the exclusion of some foods. So, future research is necessary to eventually standardize specific supplementation. Due to the slight differences between the “Bambino” diet and a standard Mediterranean diet, the long-term effects, cost, and accessibility are comparable. The continuous monitoring of patients during the routine follow-ups will provide more data on the persistence in time of the biochemical improvements, along with the improvement in clinical symptoms.

Raising awareness of X-ALD is crucial among the general population and healthcare providers and stakeholders. We strongly recommend food labeling with C26:0 content and annotation of its content, measured by a gold-standard technique, in the main food databases.

## 5. Conclusions

The “Bambino” VLCFA-restricted diet caused a dramatic improvement of the X-ALD biomarkers, along with a reduced intake of VLCFAs and fats from the diet. In the “Bambino” cohort, adherence to the diet was good, thus proving that nutritional treatment must be considered a pivotal strategy. It is crucial in presymptomatic patients and in subjects who do not have a severe form of disease to delay the onset of symptoms and to improve the prognosis [8,16,24,32]. The possibility of drawing up this nutritional pattern and all subjects receiving the same nutritional protocol and general advice, reducing the risk of bias, represent strengths of our study.

## Figures and Tables

**Figure 1 nutrients-16-03341-f001:**
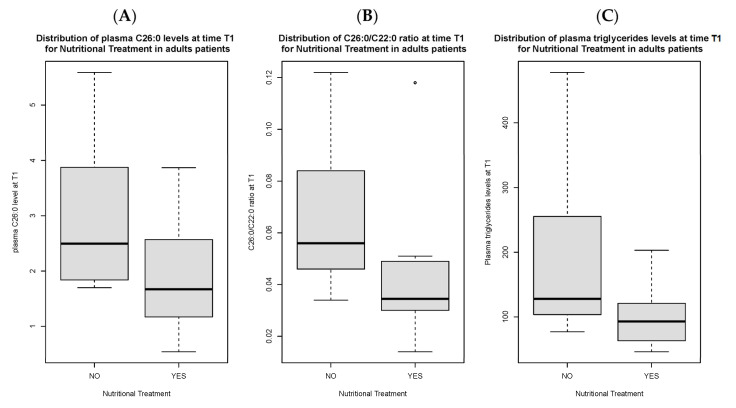
Distribution of C26:0 plasma levels (**A**), C26:0/C22:0 ratio (**B**) and triglycerides (**C**) at time T1 in adult patients compliant to nutritional treatment as compared to non-compliant patients. Figure 1B presents an outlier.

**Figure 2 nutrients-16-03341-f002:**
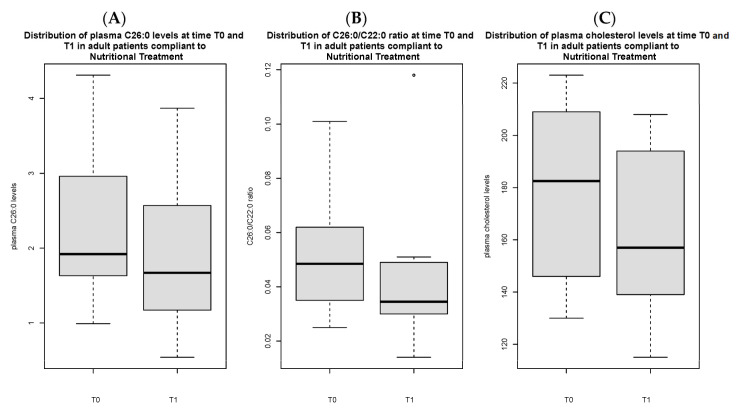
Distribution of C26:0 plasma levels (**A**), C26:0/C22:0 ratio (**B**) and cholesterol (**C**) in adult patients compliant to nutritional treatment, comparing T0 vs. T1. Figure 2B presents an outlier.

**Figure 3 nutrients-16-03341-f003:**
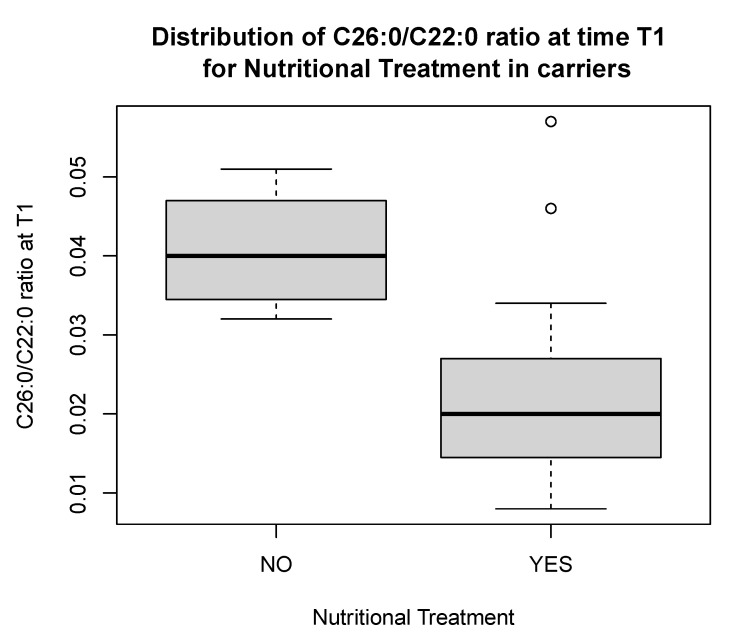
Distribution of C26:0/C22:0 ratio at time T1 in carriers compliant to nutritional treatment as compared to non-compliant carriers. The figure presents outliers.

**Figure 4 nutrients-16-03341-f004:**
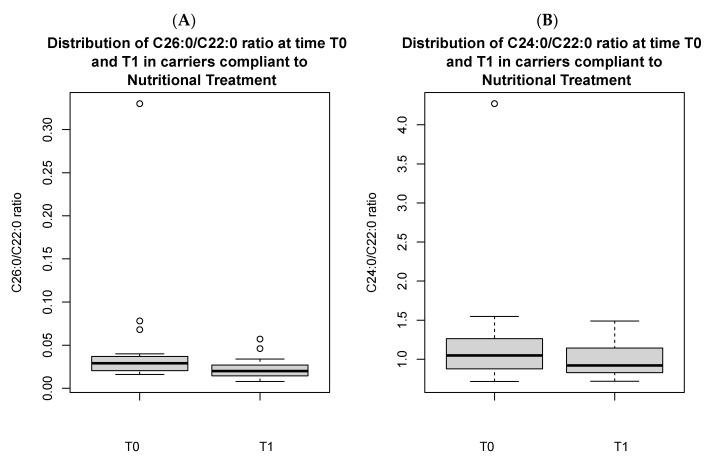
Distribution of C26:0/C22:0 ratio (**A**) and C24:0/C22:0 ratio (**B**) in carriers compliant to nutritional treatment, comparing T0 vs. T1. Both the figures present outliers.

**Table 1 nutrients-16-03341-t001:** Median and IQR [median (IQR)] of plasma VLCFA levels (C26:0, C26:0/C22:0 ratio, C24:0/C22:0 ratio), total cholesterol and triglycerides, at T0 and T1, divided by compliant and non-compliant to nutritional treatment, for each sample group.

	Patients				Carriers	
	Children		Adults		Female Carriers	
	*T0*	*T1*	*T0*	*T1*	*T0*	*T1*
Age, Years Old	8.9 (8.1)		39.1 (22.3)		51.7 (14.3)	
Weight, kg	32.6 (21)	33.3 (18.4)	72.2 (13)	72 (7.1)	61.7 (20.4)	61.6 (17)
Height, cm	139.6 (32.4)	146.3 (33.3)	173 (10.6)	173 (11.1)	158 (5.5)	158 (4.9)
BMI, kg/m^2^	17.6 (2.9)	17.3 (2.3)	23.5 (4.3)	23.8 (4.8)	24.7 (4.9)	22.3 (5.5)
C26:0, µmol/L	1.4 (0.9)	1.6 (0.7) vs. 1.5 (0.7)	2.4 (1.7)	1.7 (1.2) vs. 2.5 (1.7) * ^b^ ** ^b^	1.4 (0.7)	1.2 (0.7) vs. 1.8 (0.2)
C26:0/C22:0 ratio	0.04 (0.03)	0.03 (0.03) (A) 0.04 (0.008) (B)	0.06 (0.04)	0.04 (0.02) (A) 0.06 (0.03) (B) * ^b^ ** ^e^	0.03 (0.02)	0.02 (0.01) (A) 0.04 (0.009) (B) * ^d^ ** ^a^
C24:0/C22:0 ratio	1.2 (0.2)	1.1 (0.08) (A) 1.2 (0.2) (B)	1.4 (0.3)	1.2 (0.5) (A) 1.4 (0.4) (B)	1.0 (0.2)	0.9 (0.3) (A) 1.2 (0.3) (B) ** ^b^
Cholesterol, mg/dL	141.5 (32.8)	123 (29) (A) 149 (35) (B)	173.5 (68.3)	157 (54) (A) 193 (39) (B) ** ^b^	191 (50.8)	183 (40.5) (A) 186.5 (12.8) (B)
Triglycerides, mg/dL	63 (19.8)	62 (81) (A) 70.5 (19.5) (B)	103 (63.8)	93 (56.5) (A) 128 (109.5) (B) * ^c^	77.5 (37)	74 (48.3) (A) 84.5 (79.8) (B)
Adherence to Nutritional Treatment, %	50%		53.9%		80%	

(A): compliant to nutritional treatment; (B): non-compliant to nutritional treatment; * significant comparison between compliant and non-compliant at T1; ** significant comparison between compliant at T1 and T0. ^a^: *p* = 0.001; C 26:0: hexacosanoic acid; ^b^: *p* = 0.01; C26:0/C22:0 ratio: hexacosanoic acid/docosanoic acid ratio; ^c^: *p* = 0.02; C24:0/C22:0 ratio: tetracosanoic acid/docosanoic acid ratio; ^d^: *p* = 0.03; ^e^: *p* = 0.05.

**Table 2 nutrients-16-03341-t002:** Example of restricted Mediterranean diet vs. unrestricted one.

Restricted Diet			Unrestricted Diet		
Meal	Foods	C26:0 (mg)	Meal	Foods	C26:0 (mg)
Breakfast	200 g Skim Milk	0.1 [16]	Breakfast	150 g Ordinary Liquid Milk	0.5715 [17]
	Sandwich Made with 50 g Italian Bread and 50 g Lean Ham	0.35 (Bread) [16] + 0.036 (Ham) [16]		30 g Cereals	0.066 [16]
Snack	Fruit: 150 g Peeled and Deseed Strawberries and 150 g Watermelon	0.096 (Strawberries) [16] + 0.24 (Watermelon) [16]	Snack	150 g Peaches w/peel + 30 g Sweet Almonds	0.1365 (Peaches) [16] + 0.7548 (Almonds) [17]
Lunch	100 g rice [22]	0.7506 [16]	Lunch	80 g Rice [22]	0.5994 [16]
	200 g Skinned Chicken Breast [22]	0.108 [16]		180 g Skinned Turkey Breast [22]	0.3212 [16]
	200 g p.d. Tomatoes	0.142 [16]		200 g Lettuce Hearts	0.74 [16]
Snack	150 g Skim Milk	0.075 [16]	Snack	125 g Yoghurt	2.0125 [17]
	150 g Peeled Apples	0.0675 [16]		150 g Banana	1.17 [16]
Dinner	120 g Egg Whites	0.0084 [16]	Dinner	200 g Fresh Cod [22]	0.96 [16]
	200 g Peeled Carrots	0.122 [16]		200 g Cabbage	4.38 [16]
	60 g Italian Bread	0.42 [16]		50 g Italian Bread	0.35 [16]
Total: 2.5 mg/day C26:0; 8.7 g/day total fatty acids (5.6%) [18]; 1404 kcal/day (daily seasoning oil not included) [18]			Total: 12.1 mg/day C26:0; 33.40 g/day total fatty acids (20.0%) [18];1.501 kcal/day (daily seasoning oil not included) [18]		

## Data Availability

The data described in the study are not publicly available due to privacy and ethical restrictions but may be made available upon request.

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
