# Peer review of "Nutritional Counseling and Mediterranean Diet in Adrenoleukodystrophy: A Real-Life Experience"

_nutrients, 2024, doi:10.3390/nu16193341_

Round 1
Reviewer 1 Report
Comments and Suggestions for Authors
The manuscript presents that a Mediterranean diet restricted in Very Long Chain Fatty Acids (VLCFAs) effectively reduces plasma VLCFAs levels in patients and carriers of X-ALD, highlighting the importance of dietary compliance in managing the disorder.
Comments
1. I recommend to include an abbreviation list
2. I recommend to mention into the abstract and at Material and methods the number and gender of patients and carriers, taken into the study
3. In table 2, to remove the dot from the head of the table: Restricted diet.
4. To include a short evaluation regarding the limitations of the study (at least related to the number of the participants evaluated)
5. In suppl. material:
a. To include several examples with references for the mention ‘the majority of “Addison Only” phenotype displays neurologic symptoms during one’s life’.
b. For table 2 Supplementary, to correct - introduce a space on 2nd page, for word ‘wholemeal’... ‘whole meal’
c. For table 2 Supplementary, to correct - introduce a space on 3rd page, for word semiskimmed… semi skimmed
d. To remove the coma ‘paw paw, mango’.
Author Response
Review #1
- I recommend to include an abbreviation list.
Now included, right after Abstract and Keyword section. Lines 34-47: “Abbreviation list: ALDP, Adrenoleukodystrophy Protein; AMN, Adrenomyelomeuropathy; BDA, Banca Dati di composizione degli Alimenti; BMI, body mass index Kg/m2; CerALD, Cerebral disease ALD; CREA, Consiglio per la Ricerca in agricoltura e l’analisi dell’Economia Agraria; EHRs, Electronic Health Records; ELOVL1, ELOngation of Very Long chain fatty acids-1; LO, Lorenzo’s Oil; OPBG, Ospedale Pediatrico Bambino Gesù; USDA, U.S. Department of Agriculture; VLCFAs, Very Long Chain Fatty Acids; X-ALD, Adrenoleukodystrophy”
- I recommend to mention into the abstract and at Material and methodsthe number and gender of patients and carriers, taken into the study.
Added. Lines 19-21 “We retrospectively evaluated plasma VLCFAs in a cohort of 36 patients and 20 carriers at baseline and after 1 year of restricted diet” and lines 100-101 “We totally included 56 subjects, distributed as follows: 26 male adults, 10 male children and 20 female adults”.
- In table 2, to remove the dot from the head of the table: Restricted diet.
Done.
- To include a short evaluation regarding the limitations of the study (at least related to the number of the participants evaluated).
Thank you for the suggestion. We better explained the limits of our study. See lines 443-446 “Our study is a retrospective but real-life investigation. Ethical concern makes it very complicate to carry out randomized controlled trial in X-ALD, a rare condition [8], that also accounts for the small number of patients evaluated in the present study, despite our sample size is larger as compared to previous studies”, line 458-459 “However, more frequent in-person follow up visits could have improved compliance” and lines 461-463 “As a limit of our experience, we did not analyzed data on micronutrients intake, that could occur in patients following a restricted diet, due to exclusion of some foods. So, future research is necessary to eventually standardize specific supplementation”.
- In suppl. material:
- To include several examples with references for the mention ‘the majority of “Addison Only” phenotype displays neurologic symptoms during one’s life’.
The literature [3, 5, 6, 8, 9] confirms the scenario in which “Addison Only” phenotype may experience neurologic manifestations over the course of a lifetime.
- For table 2 Supplementary, to correct - introduce a space on 2ndpage, for word ‘wholemeal’... ‘whole meal’.
Done.
- For table 2 Supplementary, to correct - introduce a space on 3rdpage, for word semiskimmed… semi skimmed.
Done.
- To remove the coma ‘paw paw, mango’.
Done.

Reviewer 2 Report
Comments and Suggestions for Authors
Q1:Abstract: : Adrenoleukodystrophy (X-ALD) is a metabolic disorder caused by dysfunctional peroxisomal beta-oxidation of Very Long Chain Fatty Acids (VLCFAs). A VLCFAs Mediterranean restricted diet has been proposed for patients and carriers to reduce their daily intake . ?
Do you mean “Adrenoleukodystrophy (X-ALD) is a metabolic disorder caused by dysfunctional peroxisomal beta-oxidation of Very Long Chain Fatty Acids (VLCFAs). A VLCFAs Mediterranean restricted diet has been proposed for patients and carriers to reduce their daily intake of VLCFAs.”
Q2: We retrospectively evaluated plasma VLCFAs in a cohort of patients and carriers at baseline and after 1 year of diet.
Do you mean “We retrospectively evaluated plasma VLCFAs in a cohort of patients and carriers at baseline and after 1 year of VLCFAs Mediterranean restricted diet.”
Q3: line 45: AMN, please provide the full name of it.
Q4: In several places, "mcgmol/L" should be corrected to "µmol/L."
Q5:line 40: Adrenoleukodistrophy or Adrenoleukodystrophy ?
Q6: Can you briefly introduce Bambino diet in your content, and tell the readers about the differences from other Mediterranean diets.
Q7: Discussion: Consolidating these comparisons into a single paragraph might improve the flow.
Q8: Conclusion: suggest to rewrite to make it more concise and remove less critical detail and emphasize your important points.
Author Response
Review #2
Q1: Abstract: Adrenoleukodystrophy (X-ALD) is a metabolic disorder caused by dysfunctional peroxisomal beta-oxidation of Very Long Chain Fatty Acids (VLCFAs). A VLCFAs Mediterranean restricted diet has been proposed for patients and carriers to reduce their daily intake.
Do you mean “Adrenoleukodystrophy (X-ALD) is a metabolic disorder caused by dysfunctional peroxisomal beta-oxidation of Very Long Chain Fatty Acids (VLCFAs). A VLCFAs Mediterranean restricted diet has been proposed for patients and carriers to reduce their daily intake of VLCFAs”.
Yes. Added (line 19).
Q2: We retrospectively evaluated plasma VLCFAs in a cohort of patients and carriers at baseline and after 1 year of diet.
Do you mean “We retrospectively evaluated plasma VLCFAs in a cohort of patients and carriers at baseline and after 1 year of VLCFAs Mediterranean restricted diet.”
Yes. Added (line 21).
Q3: line 45: AMN, please provide the full name of it.
Done (now line 59).
Q4: In several places, "mcgmol/L" should be corrected to "µmol/L."
Done.
Q5: line 40: Adrenoleukodistrophy or Adrenoleukodystrophy?
Adrenoleukodystrophy is the correct spelling (line 53).
Q6: Can you briefly introduce Bambino diet in your content, and tell the readers about the differences from other Mediterranean diets.
Q7: Discussion: Consolidating these comparisons into a single paragraph might improve the flow.
Thank you for the suggestion. The “Bambino” diet was well explained in the Materials and Methods paragraph. Now we resumed the main differences in the first part of Discussion, so resolving Q6 and Q7. See lines 323-331 “The “Bambino” diet prescribed in our study is designed on a Mediterranean pattern, with special attention to those foods high in C26:0 and fats. Some of the pillars of the Mediterranean diet are missing in the “Bambino” diet, while keeping a great variability. Fruits, vegetables and legumes are still consumed, with just few of them not allowed, and similarly for cereals. In particular, bread, rice and white flour-derived products are included in the “Bambino” diet, although having high levels of VLCFAs; so, meeting one of the pillars of Mediterranean pattern and ensuring an adequate food variety and nutrient intake. Nevertheless, the consumption of extra virgin olive oil is excluded from the diet, being replaced by the oil specifically designed for X-ALD”.
Q8: Conclusion: suggest to rewrite to make it more concise and remove less critical detail and emphasize your important points.
We have modified the Conclusion, highlighting the strengths of our study. Lines 474-481 “The “Bambino” VLCFAs restricted diet caused the dramatic improvement of the X-ALD biomarkers, along with a reduced intake of VLCFAs and fats from the diet. In the “Bambino” cohort, adherence to the diet was good, this proving that nutritional treatment must be considered a pivotal strategy, particularly in presymptomatic patients and in subjects who do not have severe form of disease to delay the onset of symptoms and to improve the prognosis [8,16,24,31]. The possibility to draw up this nutritional pattern and all subjects receiving the same nutritional protocol and general advice, reducing risk of bias, represent the strength of our study”.

Reviewer 3 Report
Comments and Suggestions for Authors
- This research paper by Manco and colleagues studied the effects of a Mediterranean diet restricted in very long-chain fatty acids (VLCFA) on patients with X-linked adrenoleukodystrophy (X-ALD). ​ The study discussed the impact of compliance with the restricted diet on biochemical markers of X-ALD, such as plasma C26:0 levels, and highlights the differences between compliant and non-compliant individuals. ​Details on the composition of the restricted diet, food diaries showing reduced C26:0 intake, and the importance of adherence for positive outcomes are provided. ​This important study emphasizes the significance of nutritional treatment, especially for presymptomatic patients and those with milder forms of the disease, to delay symptoms and improve prognosis. ​This research addresses the need for standardized measurement methods for VLCFAs in food, the importance of early nutritional intervention through newborn screening programs, and the support from the Italian Ministry of Health for the study. ​Recommendations are made for age-specific behavioral strategies for diet compliance in children. ​The study evaluates the impact of compliance with nutritional advice on various health parameters and stresses the importance of age-specific strategies for diet compliance in children. Additionally, this study focused in detailing the challenges in standardized measurement methods for VLCFAs in food. ​ The experiments were designed and conducted appropriately and performed in accordance with suitable methods. The paragraphing is concise and easy to understand. I have the following suggestions and questions related to this work. Addressing these questions could further enrich the study findings and provide a more comprehensive understanding of the implications and applications of diet in the context of X-ALD treatment and prevention.
1. This study is interesting; however, it is essential to discuss the long-term effects on the efficacy and sustainability of the diet
2. Did the authors evaluate the detailed diet adherence information from the study groups? If yes please included,
3. Why did the authors exclude the control group?
4. There is a potential bias from the retrospective design of this study?
5. The nutrient intake data beyond C26:0 levels is ​ insufficient?
6. It might be essential to study and describe the long-term outcomes and sustainability of the diets
7. Importantly, the funding influence of the study is not explicitly talked
8. There is a need for standardized measurement methods in the design of this study.
9. Generalizability is limited to specific cohorts
10. There is a lack of complete dietary tables and nutrient deficit data ​of the study groups
11. It could be essential to describe the cost and accessibility considerations in the discussion section of the manuscript.
Author Response
Review #3
- This study is interesting; however, it is essential to discuss the long-term effects on the efficacy and sustainability of the diet.
Statistical analyses from enrolled patients show, at 1 year, a great adherence, confirming the sustainability of the “Bambino” diet (lines 456-458 “The great amount of allowed foods according to the Mediterranean pattern makes it sustainable, as proven by data on adherence revealed by statistical analysis”). Regarding long-term effects, future follow-up visits will clarify the persistence in time of the biochemical improvements and the influence on the prognosis of the disease. We better specified in lines 463-467 “Due to the slight differences between the “Bambino” diet and a standard Mediterranean diet, the long-term effects, cost and accessibility are comparable. The continuous monitoring of patients during the routine follow-ups, will provide more data on the persistence in time of the biochemical improvements, along with the improvement in clinical symptoms”.
- Did the authors evaluate the detailed diet adherence information from the study groups? If yes, please included.
Yes, we analyzed adherence in all the groups, as reported in Table 1 and in Results, lines 215-217 [“Fifty percent of the adult and child patients adhered the nutritional advice (14 out of 26 adults and 5 out of ten children), while in carriers we observed a higher percentage of adherence (N=16; 80%)”]. Children and adult patients showed 50% and 53,9% adherence, respectively. Female group was the most compliant, with an 80% of adherence. Also in lines 337-343, we discussed about the compliance to nutritional treatment “From analysis of Food Diaries and 24-h Recalls, adherence to the diet was good with almost 62.5% of participants being compliant. Among children and adults, the lowest compliance was registered, with a major consumption of non-allowed foods, especially industrial and processed items, such as sweets, meat-derived products, sweetened beverages; this was due to limits while joining parties; thus, recreational events being the major burden to follow the nutritional advice for the group of non-compliant children and adults”.
- Why did the authors exclude the control group?
The rarity of the disease makes it impossible, due to ethical concern, to have a control group, due to the lack of alternative medical therapies. Non-compliant patients served as a control group, confirming that being adherent to the diet leads to the reduction of plasma circulating levels of C26:0 and its ratios. See lines 443-444 “Ethical concern makes it very complicate to carry out randomized controlled trial in X-ALD, a rare condition [8]”.
- There is a potential bias from the retrospective design of this study?
We believe that the potential bias of the retrospective study, which involves analyzing the EHRs and food diaries a posteriori, could actually be a strength, as it represents a real-life assessment of patient compliance to the nutritional treatment, even though they were not formally part of a study.
- The nutrient intake data beyond C26:0 levels is insufficient?
There are no database referring to the food content of other VLCFAs, nor in previous studies, with C26:0 being the only one measured.
- It might be essential to study and describe the long-term outcomes and sustainability of the diets.
As for comment 1, this could be part of future research, by monitoring these patients in time.
- Importantly, the funding influence of the study is not explicitly talked.
There are no competing interests to declare. The food for special medical purpose Aldixyl (Pharmaelle) and the Aldixyl OiLife (Pharmaelle) are both reimbursed from the Italian National Health Service.
- There is a need for standardized measurement methods in the design of this study.
As our Institution is an IRCCS, methods for biochemical and anthropometric measurements are standardized, as reported in Materials and Methods section, lines 119-136 “We measured height with a wall Holtain-stadiometer and weight with scales certified for medical use (90/384/EEC, SECA, Hamburg, Germany) with patients wearing minimal clothing without shoes. The average of 2 measurements was used and the Body Mass Index (BMI) was calculated. All measures were taken to ensure the confidentiality of participants whose data were used. Blood samples were collected in EDTA tubes and plasma stored at -20°C for maximum 6 months. Plasma VLCFAs (C22:0-C26:0) were identified and quantified by gas-chromatography/mass spectrometry, after extraction and derivitization to methyl esters, using a control plasma, certified ERNDIM (European Research Network for evaluation and improvement of screening, Diagnosis and treatment of Inherited disorders of Metabolism). Plasma levels of C26:0 (hexacosanoic acid) were expressed in µmol/L; hexacosanoic to docosanoic acids ratio (C26:0/C22:0) and tetracosanoic to docosanoic acids ratio (C24:0/C22:0) were measured, since they are useful for therapeutic monitoring and for diagnosis of X-ALD in carriers. Normal range are the following: hexacosanoic acid 0.010-0.900 µmol/L, C26:0/C22:0 ratio 0.006-0.020 and C24:0/C22:0 0.470-1.27012,15. Instead, total cholesterol levels and triglycerides were assessed using colorimetric kits (modular systems P/S Can 433; Roche/Hitachi)”.
While, regarding VLCFAs food content, we expressed the need of a standardized methodology, since that previous literature is not consistent. See lines 449-453 “Moreover, some of the results about C26:0 food content, obtained by previous authors [16,17], differ, depending on techniques that have been used. So, in the perspective of further research, the need of having significant and comparable data, makes it necessary to standardize VLCFAs measurement methods, regarding analysis procedures”.
- Generalizability is limited to specific cohorts.
Our real-life investigation does not include a control group. See the answer to comment 3.
- There is a lack of complete dietary tables and nutrient deficit data of the study groups.
As a limit of our retrospective study, we did not analyzed data on micronutrients and potential deficits that could occur in patients following the “Bambino” diet. However, we provide our patients with general advice, aiming at limiting deficits, as reported in Discussion, lines 432-434 “Due to the large exclusion of foods, as general nutritional advice, we strongly recommended our patients eating fresh foods and avoiding prolonged cooking, to ensure adequate intake of all micronutrients and limit the risk of vitamin and mineral deficits”. We also declared it as a limitation. See lines 461-463 “As a limit of our experience, we did not analyzed data on micronutrients intake, that could occur in patients following a restricted diet, due to exclusion of some foods. So, future research is necessary to eventually standardize specific supplementation”.
- It could be essential to describe the cost and accessibility considerations in the discussion section of the manuscript.
Since that the “Bambino” diet does not differ from a standard Mediterranean diet, apart from the non-allowed foods, the cost and accessibility for X-ALD patients are comparable to those for general population. We also described it into the Discussion paragraph, lines 463-465 “Due to the slight differences between the “Bambino” diet and a standard Mediterranean diet, the long-term effects, cost and accessibility are comparable”. The food for special medical purpose Aldixyl (Pharmaelle) and the Aldixyl OiLife (Pharmaelle) are both reimbursed from the Italian National Health Service.

Round 2
Reviewer 2 Report
Comments and Suggestions for Authors
Accepted.
Reviewer 3 Report
Comments and Suggestions for Authors
Authors addressed the majority of concerns that I have raised and I do not have any more suggestions or revisions